# 4-OI Protects MIN6 Cells from Oxidative Stress Injury by Reducing *LDHA*-Mediated ROS Generation

**DOI:** 10.3390/biom12091236

**Published:** 2022-09-04

**Authors:** Jianmin Wu, Xingshi Gu, Juan Zhang, Ze Mi, Zhenhu He, Yuqian Dong, Wu Ge, Kedar Ghimire, Pengfei Rong, Wei Wang, Xiaoqian Ma

**Affiliations:** 1Institute for Cell Transplantation and Gene Therapy, The 3rd Xiangya Hospital of Central South University, Changsha 410000, China; 2Engineering and Technology Research Center for Xenotransplantation of Human Province, Changsha 410000, China; 3Centre for Transplant and Renal Research (CTRR), Westmead Institute for Medical Research, The University of Sydney, Sydney 2145, NSW, Australia

**Keywords:** pancreatic beta cells, itaconate, oxidative stress, lactate dehydrogenase A, hypoxia

## Abstract

Pancreatic beta cells are highly susceptible to oxidative stress, which plays a crucial role in diabetes outcomes. Progress has been slow to identify molecules that could be utilized to enhance cell survival and function under oxidative stress. Itaconate, a byproduct of the tricarboxylic acid cycle, has both anti-inflammatory and antioxidant properties. The effects of itaconate on beta cells under oxidative stress are relatively unknown. We explored the effects of 4-octyl itaconate—a cell-permeable derivative of itaconate—on MIN6 (a beta cell model) under oxidative stress conditions caused by hypoxia, along with its mechanism of action. Treatment with 4-OI reversed hypoxia-induced cell death, reduced ROS production, and inhibited cell death pathway activation and inflammatory cytokine secretion in MIN6 cells. The 4-OI treatment also suppressed lactate dehydrogenase A (LDHA)activity, which increases under hypoxia. Treatment of cells with the ROS scavenger NAC and LDHA-specific inhibitor FX-11 reproduced the beneficial effects of 4-OI on MIN6 cell viability under oxidative stress conditions, confirming its role in regulating ROS production. Conversely, overexpression of *LDHA* reduced the beneficial effects exerted by 4-OI on cells. Our findings provide a strong rationale for using 4-OI to prevent the death of MIN6 cells under oxidative stress.

## 1. Introduction

More than 451 million cases of diabetes mellitus (DM) were reported worldwide in 2017, and it is projected to reach 693 million cases by 2045 [1]. The primary pathological source of DM is insulin-producing beta cells within the pancreatic islets.

Beta-cell demise and dysfunction are increasingly recognized to play a fundamental role in the pathophysiology of type 2 diabetes (T2D). Insulin production by beta cells consumes significant amounts of oxygen to form disulfide bonds within the insulin peptide structure. Three disulfide bonds are formed to support each insulin molecule, resulting in the release of substantial amounts of reactive oxygen species (ROS) in a short time [2]. Under physiological conditions, the generated ROS activate antioxidant gene expression pathways which, in turn, inhibit ROS production and reduce oxidative stress through a negative feedback loop [3]. In addition, ROS activate multiple cellular pathways, including nicotinamide adenine dinucleotide phosphate (NADPH), which promotes calcium uptake and enhances insulin secretion from beta cells [4]. Therefore, modest levels of ROS appear to support beta-cell function.

However, in type 2 diabetic patients, beta cells utilize excessive oxygen to match higher insulin demand and compensate for insulin resistance. This process leads to hypoxia within the islet niche, resulting in excessive ROS generation and subsequent beta-cell dysfunction [5].

Due to the presence of relatively lower antioxidant enzyme activities in beta cells, they are highly susceptible to oxidative stress [6]. Moreover, the stability and functions of antioxidant enzymes are reduced during diabetes, further impairing their already malfunctional antioxidant defense and its adaptive responses to oxidative stress [7]. During hypoxia, beta cells experience oxidative stress and generate large amounts of ROS [8]. Excessive ROS can also activate inflammasomes [9] and cause necroptosis by stimulating RIPK3-mediated cell death pathways [10], and by triggering inflammation [11,12,13,14]. Furthermore, ROS can cause lipid peroxidation as well as protein and DNA damage, and are implicated in beta-cell death [15].Therefore, controlling ROS levels and enhancing the beta cells’ capacity to protect against oxidative stress are crucial in minimizing beta-cell damage and dysfunction, and will ultimately facilitate the discovery of therapeutic targets to treat diseases such as diabetes. Hypoxia-induced ROS production most likely results from the mitochondrial activity or activation of NADPH oxidase (NOX) [16]. NOX activity is reliant on the availability of NADPH [17]. Mitochondrial activity is mainly derived from the electron transfer from NADH to the oxidative respiratory chain [18]. NADH interacts with lactate dehydrogenase (LDH) to catalyze the generation of lactate from pyruvate [19]. Under hypoxic conditions, cell metabolism utilizes glycolysis, and the generation of lactate is greatly enhanced [20,21,22]. In this process, the electrons produced during the interaction of LDH and NADH enter the mitochondrial electron transport chain and are transformed into ROS, aggravating the oxidative stress state of the cells [23].

Itaconate is a byproduct of the tricarboxylic acid cycle, and is generated through cis-aconitate decarboxylation catalyzed by immune-responsive gene 1 (IRG1) in the mitochondrial matrix [24]. Initially, it was synthesized as a raw material via a chemical method [25]. Itaconate and its cell-permeable derivative 4-octyl itaconate (4-OI) have recently been shown to exhibit anti-inflammatory and antioxidant properties [26]. Uses of itaconate in diseases such as ischemia–reperfusion injury [27], obesity [28], lupus erythematosus [29], and renal fibrosis [30] have previously been reported. However, the therapeutic efficacy of itaconate or 4-OI on MIN6 cells under hypoxia is still unknown.

Inhibition of LDH-mediated ROS production could potentially inhibit MIN6 cell death under hypoxia. LDHA is a major functional subunit of LDH [23]. In this study, we provide evidence that 4-OI improves MIN6 cell survival under hypoxic conditions by reducing LDHA/NADH-mediated ROS generation, and that its use could be of therapeutic value to enhance MIN6 cell survival under stress conditions.

## 2. Materials and Methods

### 2.1. Cell Culture

The MIN6 cell line was purchased from Fenghbio Biological Ltd. (Fenghbio, Changsha, China) and tested regularly for mycoplasma contamination using a one-step mycoplasma detection kit (YiseMed, Shanghai, China). The MIN6 murine beta-cell line was cultured at 37 °C in an incubator containing 5% CO_2_ with RPMI 1640 medium supplemented with 10% FBS and 100 units/mL penicillin/streptomycin; 0.25% trypsin (Thermo Fisher Scientific, Waltham, MA, USA) was used to detach the cells, and they were passaged at a 1:3 ratio when the cell density reached around 80–90%. The hypoxia parameters were 1% O_2_ + 5% CO_2_. Various doses of 4-OI (Selleck, Houston, TX, USA), FX-11 (MCE, Monmouth Junction, South Brunswick Township, NJ, USA), and N-acetyl-l-cysteine (NAC) (Selleck, Houston, TX, USA) were used to pretreat cells for 4 h before incubation under hypoxic conditions.

### 2.2. Overexpression of LDHA

*LDHA* DNA was obtained from BALB/c mice. The PCR amplification product of the *LDHA* gene with correct sequencing was purified, and then digested by HindIII and EcoRI restriction endonuclease at 37 °C for 30 min, and the digested product was purified again, before being mixed with DNA ligase. The purified product was then ligated to a pCMVplasmid carrying a FLAG-tag (FLAG-tag is a segment consisting of 8 amino acid residues, N-DYKDDDDK-C (1012 Da), which functions as a tag), transformed into *E. coli* Trans5α receptor cells, and transformed on LB solid plates containing ampicillin. After overnight culture, the monoclonal colonies were cultured in LB liquid medium and shaken until turbid with obvious precipitation, as detected by 1.5% agarose gel electrophoresis. Then, the plasmids were extracted using a plasmid extraction kit and sent for sequencing. MIN6 cells were seeded (2 × 10^5^ per well) into 6-well plates and transfected with am *LDHA*^Flag^ plasmid using Lipofectamine™ 3000 Transfection Reagent (Thermo Fisher Scientific, Waltham, MA, USA), according to the manufacturer’s protocols.

### 2.3. Cell Viability Analysis

MIN6 cells were seeded (1 × 10^4^ per well) in 96-well plates, and the cells were cultured for 72 h under hypoxia or normoxia. Cell viability was examined using the MTS Assay Kit (Promega, Madison, NJ, USA) according to the manufacturer’s instructions.

### 2.4. Flow Cytometric Analysis

In 12-well plates, MIN6 cells were seeded at 1 × 10^5^ per well. Cells were incubated for 72 h under normoxia or hypoxia. Cells were trypsinized, and the Annexin V/PI Apoptosis Detection Kit (B&D, Franklin Lakes, NJ, USA) was used to label them. A flow cytometer was used to determine cell apoptosis, which was examined using the CXP Analysis program (Beckman Coulter, Brea, CA, USA).

### 2.5. Detection of Intracellular Reactive Oxygen Species (ROS)

The cells were seeded (2 × 10^6^ per plate) in a 10 cm dish and incubated for 72 h under either hypoxia or normoxia. The cells and their supernatants were collected after incubation. Cells were washed two times with cold PBS, followed by incubation with 10 mM DCFH-DA ( Bestbio, Shanghai, China) in the dark at 37 °C for 30 min. In the presence of ROS, DCFH was oxidized to produce the fluorescent substance DCF, and the intensity of the green fluorescence was proportional to the level of intracellular ROS. The cells were then resuspended in 500 μL PBS and examined using CXP Analysis software on a flow cytometer.

### 2.6. Enzyme-Linked Immunosorbent Assay (ELISA)

The levels of TNF-α, IL-1, and IL-6 in the cell culture supernatants after treatment were determined using ELISA assay kits (MLBIO, Shanghai, China) according to the manufacturer’s protocols. Data were normalized against protein concentration.

### 2.7. Quantitative Real-Time PCR (qRT-PCR)

Cellular mRNA was isolated using the Eastep^®^ Super Total RNA Extraction Kit (Promega, Madison, NJ, USA), and complementary DNA (cDNA) was prepared using the RevertAid First Strand cDNA Synthesis Kit ( Thermo Fisher Scientific, Waltham, MA, USA), according to the manufacturers’ protocols. Quantitative PCR was performed using Power Up™ SYBR™ Green Master Mix (Thermo Fisher Scientific, Waltham, MA, USA) in an Applied Biosystems 7500/7500 Fast Real-Time PCR System ( Thermo Fisher Scientific, Waltham, MA, USA) with the following conditions: 40 cycles of 50 °C for 2 min, 95 °C for 2 min, 95 °C for 15 s, and 60 °C for 60 s. All primers are recorded in Appendix A.

### 2.8. LDH Enzyme Activity Assay

MIN6 cells were plated in 12-well plates for 72 h, and the supernatants were collected. For recombinant LDHA protein, the protein was co-incubated with 4-OI or FX-11 at 37 °C for 1 h, and the solution was collected. Assays were performed on 96-well plates following the protocol of the LDH Activity Assay Kit (Solarbio, Shanghai, China) to assess the enzyme activity of LDH or LDHA. Absorbance measurements were performed using a microplate reader, and data were analyzed using Gen5 software.

### 2.9. Measurement of NADPH Levels

The Glutathione Peroxidase Assay Kit (Abcam, Cambridge, UK) was used to measure NADPH levels in the cell culture supernatants, according to the manufacturer’s instructions. Protein concentration was used to normalize NADPH levels. Gen5 software was used to collect and process the data.

### 2.10. Measurement of NADH Levels

The levels of NADH in the cell culture supernatants were assessed using the Glyceraldehyde 3 Phosphate Dehydrogenase Activity Assay Kit (Abcam, Cambridge, UK) according to the manufacturer’s protocols. The levels of NADH were normalized against protein concentration. Data were collected and processed using Gen5 software.

### 2.11. Western Blotting

Cells were washed once with cold PBS, centrifuged at 3000× rpm for 2 min, and 200 μL RIPA (Thermo Fisher Scientific, Waltham, MA, USA) was added to the lysate. The solution after lysate was sonicated for 1.5 min, and protein was extracted. The BCA Protein Assay kit (Thermo Fisher Scientific, Waltham, MA, USA) was used to assess the total protein levels. The protein was loaded in polyacrylamide gels in amounts ranging from 20 to 30 mg. Antibodies were used after transferring the proteins to a PVDF membrane. HRP-conjugated secondary antibodies were used to detect protein expression using chemiluminescence. ImageJ software was used to perform densitometry on scanned immunoblot images. The band density for the target proteins was normalized against β-actin to obtain a relative density. All antibodies are recorded in Appendix A.

### 2.12. Confocal Microscopic Analysis

MIN6 cells were grown in a special glass-bottomed plate (Biosharp, Shanghai, China) used for confocal microscopy imaging. DCFH-DA and Hoechst 33324 were then used to label the cells according to the manufacturer’s instructions. Stained cells were observed using an Olympus FV1000 confocal laser scanning microscope. Green fluorescence showed ROS formation staining with DCFH-DA, while blue fluorescence showed nuclear staining with Hoechst33342.

### 2.13. Acridine Orange (AO)/Propidium Iodide (PI) Staining

AO/PI staining was used to detect the viability of MIN6 cells after the corresponding treatments. Cells were stained with ViaStain™ AO/PI Staining Solutions (Nexcelom, Boston, MA, USA) and viewed under a fluorescence microscope (Olympus, Tokyo, Japan). Red fluorescence indicated dead cells, while green indicated living cells.

### 2.14. Mass Spectrometry

Recombinant LDHA protein (SB, 51207-M07E) was co-incubated with 4-OI or FX-11 at 37 °C for 1 h, and the solutions were collected. Protein concentration was determined using the BCA method, and 2 μg of each sample was subjected to SDS-PAGE.

For enzymatic desalting of the gum strips, the samples were desalted using a C18 desalting column. Then, 100% acetonitrile was used to activate the desalting column, and 0.1% formic acid was used to equilibrate the column. Samples were loaded onto the column, followed by washing with 0.1% formic acid (Sigma Aldrich, Burlington, NJ, USA) to clear away impurities. Finally, 70% acetonitrile was used for elution, where the flow-through solution was collected and lyophilized. Mobile phases A (100% water, 0.1% formic acid) and B (80% acetonitrile, 0.1% formic acid) were prepared. Lyophilized powder was dissolved with 10 µL of liquid A, centrifuged at 14,000× *g* for 20 min at 4 °C, and 1 µg of supernatant was extracted from the sample to be tested. An Orbitrap Exploris™ 480 mass spectrometer with an optional FAIMS Pro™ Interface and a Nanospray Flex™ (NSI) ion source was used to generate raw data (.raw) for mass spectrometry detection. The resulting MS/MS data were processed using Proteome Discoverer 2.4. Tandem mass spectra were searched against the UniProt database. The dynamic modification was set as C5H6O4(C). Trypsin was specified as a cleavage enzyme. The mass error was set to 15 ppm for precursor ions and 0.02 Da for fragment ions.

### 2.15. Statistical Analyses

Data are presented as the mean ± SD (unless otherwise indicated) of the results from at least 3 independent cell cultures. According to the type of data, comparisons were made using an unpaired Student’s *t*-test, one-way or two-way Analysis of Variance (ANOVA), and Tukey’s test for multiple comparisons. Statistical significance was established at the levels of * *p* < 0.05, ** *p* < 0.01, *** *p* < 0.001, and **** *p* < 0.0001, where *p* < 0.05 was considered statistically significant. GraphPad Prism8 was used for statistical analysis (GraphPad Software Inc., San Diego, CA, USA).

## 3. Results

### 3.1. 4-OI Enhances the Viability of MIN6 Cells under Hypoxic Conditions

We initially examined the effects of hypoxia on MIN6 cell survival. The MIN6 cell line was chosen for in vitro studies because it is a widely used beta-cell line of murine origin that has been experimentally validated to represent key physiological processes, including insulin secretion, glycolysis, and oxidative phosphorylation [31]. MIN6 cells were cultured for different durations under hypoxic conditions. They were then stained with propidium iodide (PI) and Annexin V—two reliable markers of cell death. The viability of MIN6 cells cultured under normoxia was above 90%, and their proliferation was normal (Appendix A). Our FACS data showed that the PI- and Annexin-V-negative fraction of cells gradually decreased as the duration of hypoxia increased (Figure 1A,B). Since cell viability was extremely low after 96 h of hypoxia, we decided to choose a 72 h incubation period to explore the effects of 4-OI on MIN6 cells for subsequent experiments. Hypoxia increased dichloro-dihydro-fluorescein diacetate (DCFH-DA) incorporation, which indicated elevated ROS activity within MIN6 cells (Figure 1C,D) as well as increased HIF1-α expression, significantly validating our experimental model (Figure 1E,F). We then examined the effects of 4-OI on MIN6 cells under normoxia. The 4-OI treatment did not affect cell viability under normoxia at 24 h (Appendix A), but at high concentrations (250–500 μM) it significantly boosted cell viability at 72 h (Appendix A). The cells were then treated with various doses of 4-OI for 72 h under hypoxic conditions. Hypoxia alone was sufficient to reduce MIN6 cell viability by almost 40%. This could be prevented with 4-OI treatment in a dose-dependent (62.5–250 μM) manner, with the strongest effect seen at a 125 μM 4-OI, while 500 μM appeared to be toxic to cells (Figure 1G). The enhanced MIN6 cell viability obtained with 125 μM 4-OI treatment under hypoxic conditions was validated by FACS (Figure 1H,I). These findings indicate that hypoxia causes MIN6 cell death, and that this could be substantially prevented with 4-OI treatment.

### 3.2. 4-OI Inhibits Hypoxia-Induced Cell Death by Reducing ROS Production

Hypoxia activates oxidative stress to trigger cell death [5]. To explore the mechanism of the protective effect of 4-OI, we analyzed the redox state of cells under hypoxia. MIN6 cells were treated with 5 mM NAC—an inhibitor of ROS [32]—and placed under hypoxia. ROS production in cells was investigated with DCFH-DA staining. Our confocal microscopy results indicated that hypoxia caused an imbalance in oxidative homeostasis by stimulating the production of intracellular ROS. The addition of 4-OI or NAC significantly reduced the contents of cellular ROS in all three groups of hypoxic cells (Figure 2A,B). These results were also confirmed by FACS (Figure 2C,D). Furthermore, 4-OI treatment provided comparable protective effects to NAC in enhancing cell viability when compared to the controls (Figure 2E,F). These results show that 4-OI protects MIN6 cells during hypoxia by inhibiting the production of ROS.

### 3.3. 4-OI Alkylates LDHA, Leading to Reduced ROS Generation

NADPH and NADH represent the two main pathways for intracellular ROS production during hypoxia. NADPH did not change considerably before or after hypoxia. However, we found that hypoxia significantly increased the contents of NADH (Appendix A) and transcript levels of *LDHA* (Appendix A). This suggests that the ROS produced in MIN6 cells during hypoxia are mainly of NADH origin.

NADH interacts with LDH to generate electrons, and promotes ROS production. LDH consists of LDHA and LDHB subunits. LDHA is the main functional subunit of LDH. Moreover, 4-OI has been reported to alkylate the thiol in cysteine residues of LDHA through the Michael addition [26]. To further clarify the working mechanism of 4-OI in reducing ROS generation, mass spectrometry was performed to examine whether recombinant LDHA was alkylated by 4-OI. Our results showed that Cys113 of LDHA was alkylated by itaconate (Figure 3A,B). Previously, 4-OI has been shown to hydrolyze to itaconate at 37 °C [33], and this was supported by our results (Appendix A). Next, we co-incubated the recombinant LDHA protein with 4-OI or FX-11—a specific inhibitor of LDHA in vitro—and in both cases, the enzymatic activities of LDHA were decreased in a dose-dependent manner (Figure 3C,D). We treated MIN6 cells under hypoxia with FX-11 at different concentrations. Our results revealed that 25 μM FX-11 had the highest protective effect under hypoxia, which could be due to the stress damage caused by high-dose medication (Appendix A). We examined the enzymatic activity of intracellular LDH after 4-OI or FX-11 treatments. We found that the activity of LDH was increased significantly under hypoxia, while treatment with 4-OI or FX-11 significantly downregulated it (Figure 3E). We further evaluated the effects of 4-OI, FX-11, and *LDHA*^Flag^ transfection on ROS generation under hypoxia. Quantitative PCR and Western blotting results demonstrated that our transient transfection of LDHA was successful (Appendix A). The treatment with 4-OI exhibited similar effects to FX-11 in reducing intracellular ROS, and overexpression of *LDHA*^Flag^ in MIN6 cells reversed the inhibitory effect of 4-OI on ROS (Figure 3F,G). Viability was measured via AO/PI cell staining. Both 4-OI and FX-11 significantly improved the MIN6 cell viability under hypoxia at 72 h, and the transfection of LDHA^Flag^ reversed this improvement (Figure 3H,I)). These findings indicate that 4-OI protects MIN6 cells from hypoxia-induced damage by reducing LDH activity.

### 3.4. 4-OI Reduces Cell-Death-Related Protein Expression in MIN6 Cells under Oxidative Stress

High ROS production can activate multiple cell death pathways. Several key proteins are involved in the cell death pathways. The expression of cell-death-associated marker proteins was examined under normoxia and hypoxia after 4-OI treatment. The apoptosis marker protein caspase-3, as well as the key necrosis-related proteins RIPK-3 and MLKL [34], all showed enhanced expression in response to hypoxia (Figure 4). The expression of the key autophagy-related protein LC3-II/I was decreased under hypoxia. These results suggest that hypoxia initiates the MIN6 cell death process. In contrast, the expression of these proteins was significantly reduced when the cells were treated with 4-OI, NAC, or FX-11—even when they were under hypoxia (Figure 4). Treatment with 4-OI, the scavenging of ROS, or the inhibition of LDHA activity reduced the activation of cell death pathways and inhibited the cell death process.

### 3.5. 4-OI Reduces the Release of Inflammatory Cytokines in MIN6 Cells under Oxidative Stress

ROS generation damages cells through the release of inflammatory factors. We measured the levels of inflammatory cytokines released from MIN6 cells cultured under hypoxia for 72 h at the mRNA and protein levels using qPCR and WB, respectively. Hypoxia greatly increased the levels of IL-1β, IL-6, and TNF-α released from MIN6 cells. Treatment of these cells with 4-OI, NAC, and FX-11 showed reductions in *IL-1β*, *IL-6*, and *TNF-α* expression when compared to the untreated controls by qPCR (Figure 5A–C) and WB (Figure 5D–F). These results suggest that 4-OI treatment minimizes hypoxia-induced damage in MIN6 cells in several ways, such as by reducing the release of inflammatory factors upon increased ROS, and by blocking the interaction between LDH and NADH, which reduces the production of ROS.

## 4. Discussion

Failure to secrete enough insulin by beta cells is a pathological feature of type 2 diabetes mellitus (T2DM). In T2DM, excess insulin production by beta cells to compensate for insulin resistance in peripheral tissues causes high oxygen consumption, resulting in hypoxia within and around the beta cells [35]. Beta cells undergo oxidative stress and produce considerable amounts of ROS when exposed to hypoxia. However, beta cells are particularly susceptible to oxidative stress damage [36,37]. Thus, protection of beta cells from hypoxia-induced oxidative stress damage remains an important objective that has not yet been addressed to develop treatment strategies for T2DM.

In this study, we utilized the MIN6 pancreatic beta cell line to study the effects of 4-OI—an itaconate derivative—in minimizing oxidative stress-induced damage. MIN6 cells exhibit characteristics of glucose metabolism and glucose-stimulated insulin secretion similar to those of normal islets [38]. We confirmed that MIN6 cell viability decreases with the increase in the duration of hypoxia. More than 50% of the cells died after 48 h in culture under hypoxia, and a 96 h incubation led to the demise of almost all of the cells.

The 4-OI treatment functions by alkylating different proteins to trigger downstream immunomodulatory and anti-inflammatory pathways [39,40,41]. As an unsaturated dicarboxylic acid, 4-OI alkylates cysteine residues on various proteins, such as glyceraldehyde 3-phosphate dehydrogenase (GAPDH) [42], succinate dehydrogenase (SDH) [43], and NOD-like receptor protein 3 (NLRP3) [44], and affects the corresponding downstream signals. For instance, 4-OI limits NLRP3-NEK7 interaction by alkylating C548 on NLRP3 to block the release of IL-1β and IL-18 [45]. Here, we used 4-OI treatment on MIN6 cells under hypoxia to observe its functional effects. We found that 4-OI (125 μM) treatment of MIN6 cells reversed the decline in cell viability caused by hypoxia.

Pancreatic beta cells are among the most metabolically active cells in the human body, and they are highly dependent on oxidative metabolism for energy synthesis—especially at high glucose concentrations [46]. Although islets where beta cells reside only occupy 1–2% of the whole volume of the pancreas, they receive up to 15% of the pancreatic blood supply [47]. Beta cells are highly susceptible to hypoxia, which leads to oxidative stress through ROS production. Oxidative stress increases the release of inflammatory factors and activates the cell death pathways [48]. Moreover, ROS can activate the body’s immunological response, including lymphocyte and macrophage activation and infiltration [49,50]. Treatment of Zucker diabetic fatty rats with the antioxidant NAC prevented hyperglycemia, glucose intolerance, and defective insulin secretion [51]. Similarly, treatment of diabetic db/db mice with the antioxidant agent astaxanthin helped to maintain glucose-stimulated insulin secretion, and resolved chronic inflammation [52]. The antioxidant effect of 4-OI has been reported in previous studies. By reducing oxidative stress and inflammation, 4-OI inhibited LPS-induced acute lung injury in mice, which was confirmed by the reduced lung tissue damage and inflammatory response [53]. Furthermore, 4-OI administered through tail-vein injection attenuated UVB-induced skin damage in mice by inhibiting ROS production [54]. Since ROS are one of the main factors driving islet cell death during hypoxia, we hypothesized that 4-OI could protect MIN6 cells against hypoxic injury through the inhibition of ROS production. Indeed, when MIN6 cells under hypoxia were treated with 4-OI, their cell viability improved, and the quantity of intracellular ROS was reduced.

The most likely source of intracellular ROS is mitochondrial activity or NADPH oxidase (NOX). NOX activity is reliant on NADPH, which is produced by pentose phosphate activity [55]. Mitochondrial activity is mainly achieved through the electron transfer from NADH to the oxidative respiratory chain [56]. We demonstrated that NADPH levels did not significantly change after hypoxia in MIN6 cells, whereas NADH was dramatically increased. This suggests that the ROS generated to trigger MIN6 cell death originated from the interaction of NADH and LDH. In addition, the enzymatic activity of LDHA is closely related to the viability of MIN6 cells under hypoxia. Previously published findings showed that 4-OI inhibits LDH activity by alkylating LDHA [57]. However, the inhibition of LDH activity caused by 4-OI did not affect the cellular activity and function of MIN6 under physiological conditions [58]. In this study, we found that 4-OI could modify LDHA and decrease LDH activity under hypoxia. To explore whether the inhibition of LDHA activity could rescue the cells under hypoxia, we treated hypoxic cells with FX-11—a specific inhibitor of LDHA—and found that it enhanced cell survival and inhibited ROS production in MIN6 cells. Its protective effects on MIN6 cells under hypoxia were comparable to those of 4-OI treatment.

In addition, 4-OI, NAC, and FX-11 reduced the levels of inflammatory cytokines, including IL-6 and TNF-α, and limited the activation of cell death pathways. Our Western blotting results show that hypoxia decreased the expression of key autophagy-related proteins, while the treatment with 4-OI, NAC, or FX-11 reversed this trend, suggesting that autophagy may play a protective role in limiting hypoxic injury in pancreatic islet cells. This is also consistent with our previous findings that autophagy can inhibit the hypoxia-induced death of neonatal porcine islet cells in MSC-conditioned media [59]. In addition, hypoxia stimulated the expression of key apoptotic proteins and proteins associated with necrosis pathways, where the latter were showed a greater increase. This suggests that islet cell death under hypoxic conditions is predominantly necrotic. The results of our flow cytometric apoptosis staining are also consistent with this inference, with the dead cells mainly being PI single-positive and less AV-positive. Inhibition of the necrotic pathway could be a potential strategy to inhibit necrosis of hypoxic islet cells, which is beyond the scope of this manuscript, but could be examined in future studies.

Pancreatic diseases are spreading at an exponential rate, putting a strain on healthcare systems around the world [60,61,62]. Based on the 2020 National Diabetes Statistics study from the Centers for Disease Control and Prevention, there are currently about 13% diabetic and 34.5% prediabetic adults in the US alone [63]. Oxidative stress plays an important role in the development of diabetes; 4-OI can inhibit oxidative stress damage to pancreatic islet cells, and could be an attractive treatment option for diabetes, or for prediabetes which, in turn, would inhibit the progression of diabetes. Taken together, our findings show for the first time that 4-OI can protect MIN6 cells against hypoxia-induced oxidative stress damage by reducing ROS production. We also demonstrated that 4-OI suppresses ROS-mediated hypoxic damage by limiting the activity of LDHA (Figure 6). Reducing ROS generation could be a viable option to improve outcomes of diabetes and other oxidative-stress-associated pancreatic diseases during clinical therapy.

## Figures and Tables

**Figure 1 biomolecules-12-01236-f001:**
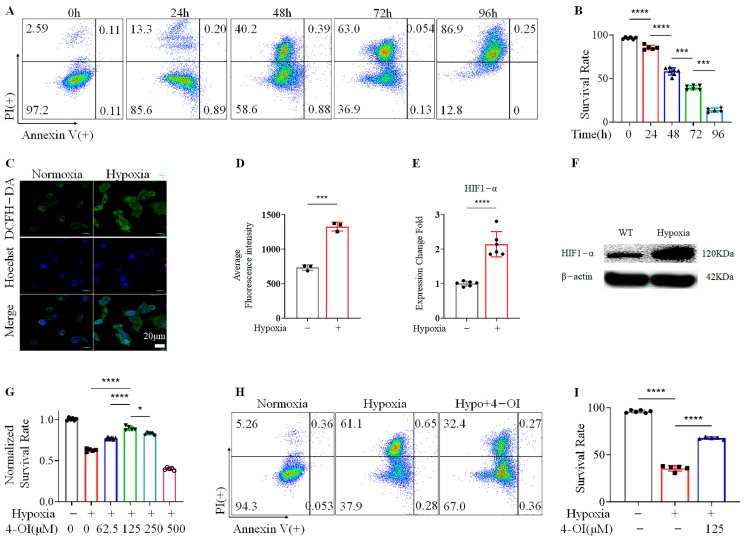
Treatment with 4-OI inhibits MIN6 cell death under hypoxic conditions: (**A**,**B**) FACS analysis showing MIN6 cell viability (stained with PI and Annexin V) under various durations of hypoxia (**A**), and its quantification (**B**). In FACS, the horizontal axis indicates positive Annexin V staining, and the vertical axis indicates positive PI staining. (**C**,**D**) ROS levels as assessed by confocal microscopy in MIN6 cells under hypoxia (**C**), and their quantification (**D**). (**E**) HIF1-α gene expression was assessed by quantitative real-time PCR in cells cultured under hypoxia for 72 h. (**F**) The expression of HIF1-α in MIN6 cells under hypoxia for 72 h was verified by Western blotting. (**G**) The effects of various concentrations of 4-OI on MIN6 cell viability when cultured under hypoxia for 72 h were measured by MTS. (**H**,**I**) FACS analysis showing the protective effect of 4-OI on MIN6 cells at 125 μM concentration (**H**), and its quantification (**I**). Error bars represent the mean ± SD. An unpaired *t*-test (**D**,**E**) or one-way ANOVA (**B**,**G**,**I**) was performed with * *p* < 0.05, *** *p* < 0.0005, **** *p* < 0.00005, ns: non-significant.

**Figure 2 biomolecules-12-01236-f002:**
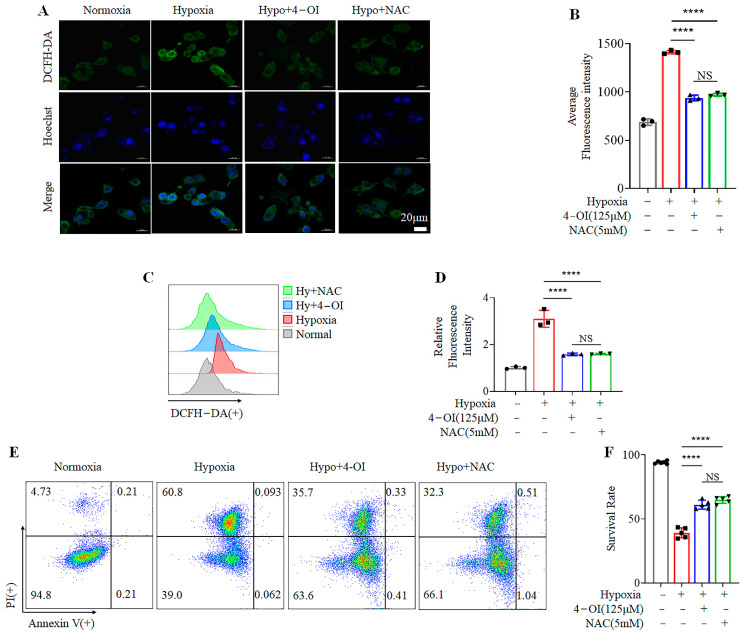
Treatment with 4-OI protects MIN6 cells from hypoxia by reducing ROS production: (**A**,**B**) ROS levels assessed by confocal microscopy in MIN6 cells after 4-OI or NAC treatment under hypoxia (**A**), and their quantification (**B**). (**C**,**D**) FACS results showing reduced ROS levels after the addition of 4-OI or NAC to cells (**C**), and their quantification (**D**). In FACS, more displacement on the horizontal axis indicates more positive ROS staining. (**E**,**F**) Survival rates of MIN6 cells treated with 4-OI or NAC under hypoxia measured with FACS (**E**), and their quantification (**F**). Error bars represent the mean ± SD. One-way ANOVA was performed with **** *p* < 0.00005, NS: non-significant.

**Figure 3 biomolecules-12-01236-f003:**
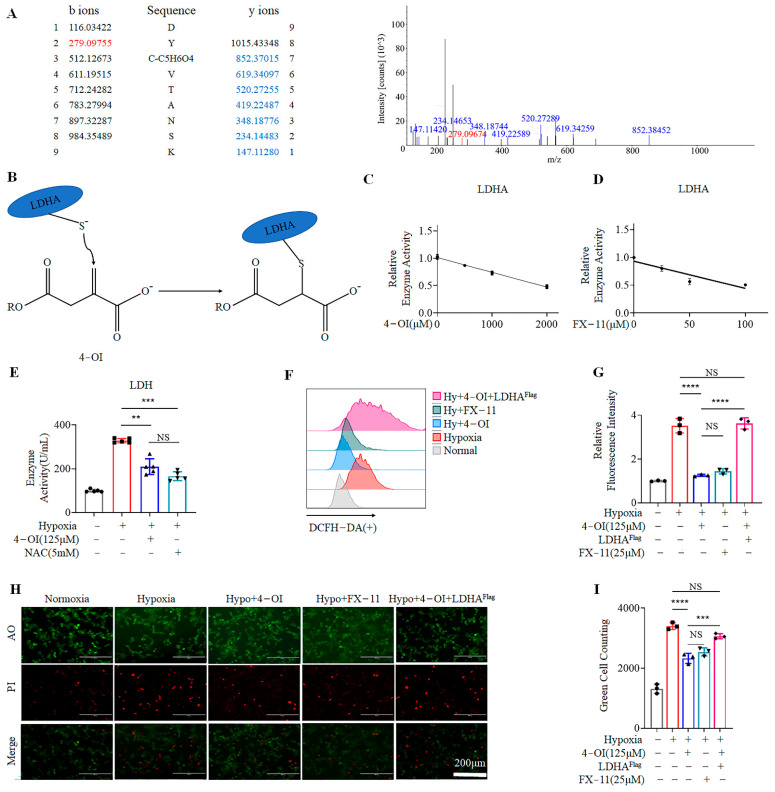
Treatment with 4-OI impairs ROS generation by modifying LDHA, and inhibits LDH activity to minimize hypoxia damage: (**A**) Mass spectrometry analysis used to identify the alkylation of LDHA caused by 4-OI. (**B**) Schematic diagram of the GAPDH thiol group covalently modified by 4-OI. (**C**,**D**) Enzymatic activity results for LDHA treated with 4-OI or FX-11. (**E**) The LDH enzyme activity of MIN6 cells after treatments was measured. (**F**,**G**) The levels of ROS and relative fluorescence intensity in MIN6 cells when treated with 125 μΜ 4-OI, 25 μΜ FX-11, or overexpression of *LDHA*^Flag^, was assessed by FACS (**F**), and their quantification (**G**). (**H**,**I**) Survival rates of MIN6 cells cultured under hypoxia for 72 h examined by AO/PI staining (**H**), and their counts of green-stained cell (**I**). Error bars represent the mean ± SD. One-way ANOVA was performed with ** *p* < 0.005, *** *p* < 0.0005, **** *p* < 0.00005, NS: non-significant.

**Figure 4 biomolecules-12-01236-f004:**
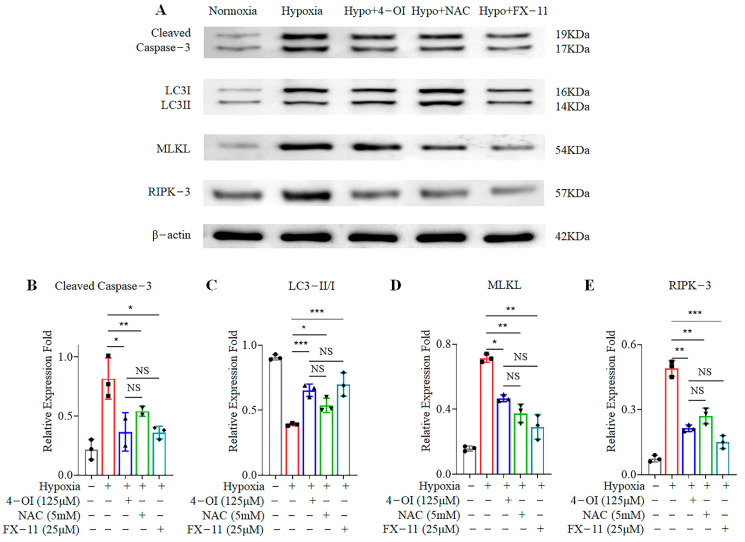
Treatment with 4-OI limits the activation of cell death pathways in MIN6 cells under oxidative stress: (**A**) The expression of key cell-death-related proteins in MIN6 cells with 4-OI or NAC or FX-11 treatments was determined by Western blotting. (**B**–**E**) The quantified graphs of the gray scale of the proteins in Western blots. Error bars represent the mean ± SD. One-way ANOVA was performed with * *p* < 0.05, ** *p* < 0.005, *** *p* < 0.0005, NS: non-significant.

**Figure 5 biomolecules-12-01236-f005:**
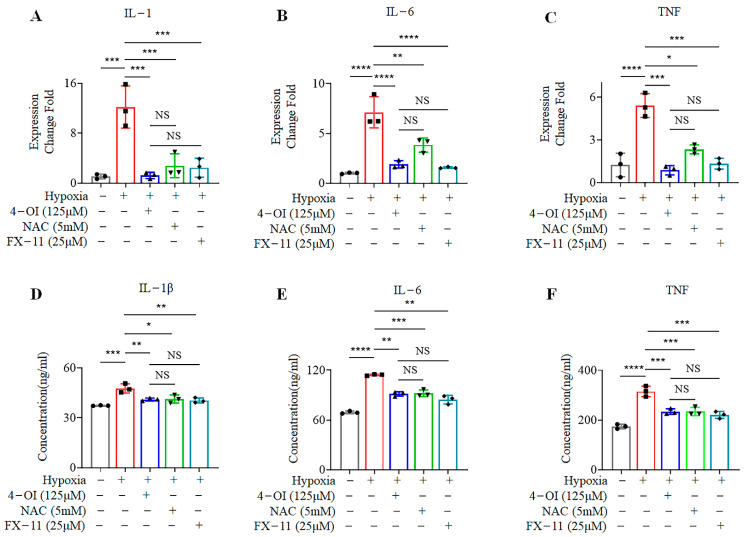
Treatment with 4-OI reduces the secretion of inflammatory factors in MIN6 cells under oxidative stress: (**A**–**C**) The mRNA levels of *IL-1*, *IL-6*, and *TNF-α*, respectively, in MIN6 cells after the indicated treatments, quantified by q-PCR. (**D**–**F**) The released levels of IL-1β, IL-6, and TNF-α, respectively, from MIN6 cells under hypoxia was assessed by ELISA. Error bars represent the mean ± SD. One-way ANOVA was performed with * *p* < 0.05, ** *p* < 0.005, *** *p* < 0.0005, **** *p* < 0.00005, NS: non-significant.

**Figure 6 biomolecules-12-01236-f006:**
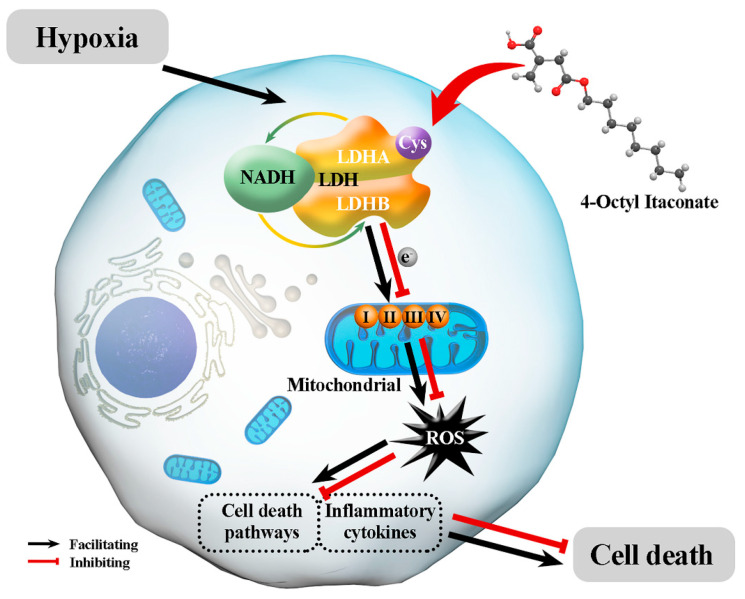
A proposed mechanism for the protective effects of 4-OI on MIN6 cells damaged by hypoxia and oxidative stress: Hypoxia promotes the interaction of NADH and LDH to generate electrons, which pass through the mitochondrial oxidative respiratory chain complex I, II, III, IV to produce reactive oxygen species. ROS activates multiple cell death pathways and increases the release of inflammatory factors, ultimately leading to cell death. Treatment with 4-OI decreases the enzymatic activity of LDH by alkylating LDHA and reducing ROS generation, thereby reducing oxidative stress and minimizing hypoxic damage.

## Data Availability

The data used to support the conclusions of this study can be obtained from the corresponding author upon request.

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
