# Peer review of "4-OI Protects MIN6 Cells from Oxidative Stress Injury by Reducing LDHA-Mediated ROS Generation"

_biomolecules, 2022, doi:10.3390/biom12091236_

Round 1

Reviewer 1 Report

The insulin-producing beta cells are highly sensitive for oxidative stress. In this manuscript, Wu et al. investigate if 4-OI, an itaconate derivate, protects beta cells from oxidative stress caused by hypoxia and determine its mechanism of action.  

Major comments:

·      The introduction is vague and not well structured. The introduction on oxygen consumption / oxidative stress is difficult to follow, for example it is not made clear how the beta cells normally modulate the generation of oxygen species. The authors start with Type 1 and Type 2 Diabetes, but only for Type 1 Diabetes it is explained in some words how protecting against oxidative stress can be an advantage for the disease. Since the special issue focusses on Type 2, more information on Type 2 would be beneficial. In line 70 the authors state that itaconate has been used previously in other diseases, but that the efficacy of 4-OI on beta cells is still unknown. Does this mean that itaconate itself has been tested in context of diabetes and the novelty here lies within the derivate?

·      It is not clear which statistical test is used for which part of the study; it should be at least mentioned in the figure legends which specific test was used. 

·      The description of the results depicted in Figure 1 should explain the difference between C, D, and E. It is not clear that the survival rate in C is measured with another assay than D and E and why are only three of the six groups shown in D and E?

·      The n for each assay should be mentioned in the figure legends. 

·      Why where much higher 4-OI concentrations used for Figure 3B compared to the previous figures? In this way it seems not relevant to the other experiments. It is also not clear which concentration of 4-OI and FX-11 are used for 3D/F and H. 

·      Why do figure 3F and H not include a LDHAflag condition treated with FX-11? The authors state in the main text (line 265 – 267): The treatment of 4-OI exhibited similar effects with FX-11 in reducing intracellular ROS and the transfection of LDHAflag increasing the level of ROS in cells. This seems to suggest that they also have the same effect with the LDHAflag cells, but that is not the case. This should be described more clearly. 

·      In line 286 – 287 the authors state that the protein expression of LC3 was decreased under hypoxia, but when you look at Figure 4A, the expression of LC3 seems increased during 24 or 72 hours of hypoxia compared to normoxia. Can the authors explain this?

·      The difference between normoxia and hypoxia never seems to be significant (or at least it is not indicated as significant), is this correct? For figure 5D it would be important to know if the difference between normoxia and hypoxia is significant, otherwise there is nothing to ‘treat’ with the 4-OI. 

·      Like in the introduction, the reasoning behind why it is important to prevent oxidative stress in beta cells is difficult to follow in the discussion. This section should be rewritten. 

·      It should be made clearer what the exact novelty of this study is compared to other studies using 4-OI, which are described in the discussion. 

·      In the discussion the authors refer to islet cells several time, but these experiments are done with a mouse insulinoma cell line. Additional experiments with (human) islets would highly increase the impact of this manuscript. 

·      The authors make use of pre-treatment of cells before hypoxia, is this relevant for clinical disease conditions? A discussion on how this could be applied in the clinic should be made clearer in the discussion. 

·      A grammar/sentence structure check is recommended, especially for the introduction. 

Minor comments:

·      Complex title – a more simplified title with less abbreviations would make it more attractive. 

·      The abbreviation of reactive oxygen species is given in line 52, while this abbreviation is already used in line 39. The abbreviation ROS is also not explained in the abstract. Also, the abbreviation 4-OI is not explained in the abstract. It is recommended to check if all the abbreviations are used correctly. 

·      In line 82, the authors mention a xxx kit, is this really the name of the kit they used for the mycoplasma tests?

·      In line 158, the authors state that cells were grown in glass bottom plate for confocal microscopic analysis. But are harvested by trypsinization. So why are they cultured on glass if they are harvested before analysis?

·     Why is the figure legend of Figure 6 in bold? Please make this consistent with the other figures. In addition, figure 6 is not mentioned in the main text, please refer to figure 6 in the corresponding text.

Reviewer 2 Report

In this manuscript, Wu et al revealed mechanisms underlying for the role of 4-OI in protecting beta cell death from oxidative stress and hypoxic conditions. They found 4-OI treatment reversed hypoxia-induced beta cell death by decreasing ROS production. In hypoxia condition, treatment on Beta cells with NAC and LDHA-specific inhibitor FX-11 reproduced the beneficial effects of 4-OI on beta cells viability, moreover, LDHA overexpression reduced the beneficial effects exerted by 4-OI on beta cells. In summary, 4-OI plays protective role in ROS-induced beta cell death by regulating activity of LDH. This research is interesting as it extends knowledge on hypoxia-induced oxidative damage occurred in T1D and T2D. Authors used appropriate cell methods to support their hypothesis. But some limitations exist, and the authors should consider addressing them to strengthen the manuscript.

Major:

1.      Authors created the hypoxia model on min6 cells, which kind of hypoxia model was used? What the concentration of O2 during the process? Authors should describe the detail of the hypoxia model in method.

2.       To detect the efficiency of hypoxia model, HIF-1 alpha and ROS expression should be detected.

3.       In figure 1, min6 viability under various duration of hypoxia was detected, the viability of min6 cells under normal oxygen concentration under the same timepoint also needs to be detected. What about the min6 proliferation under hypoxia condition?

4.      In figure 4, cleaved-caspase 3 was detected as the cell death marker, cleaved-caspase 3 is also the function molecule in apoptosis, which kind of apoptosis occurred during this process? (Intrinsic or extrinsic?) What about apoptosis receptors (Fas, DR5, and TNFR) expression with or without 4-OH treatment?

5.      In figure 6, 4-OH treatment reduced IL-1, IL-6, TNF, and IL-1beta expression, what about NF-kB and JNK cascades activity during this process?

Minor:

1.      The quality of figure 3G should be improved, some red singles are not merged with green singles.

2.       Some group have published data about the relationship between hypoxia and inflammatory factors expression, these references should be discussed. (PMID:28073079, 29526538, 21323543)

Reviewer 3 Report

In this manuscript, the authors demonstrate that 4-OI, a cell-permeable derivative of itaconate, is able to protect MIN6 cells from a hypoxia-induced oxidative stress injury. Furthermore, the authors go on to show that 4-OI treatment reversed hypoxia-induced beta cell death, reduced ROS production, inhibited cell death pathway activation, and inflammatory cytokine secretion in beta cells. 4-OI also suppressed LDHA activity, which increases upon hypoxia. Treatment of beta cells with ROS scavenger N-acetylcysteine (NAC) and LDHA-specific inhibitor FX-11 reproduced the beneficial effects of 4-OI on beta-cell viability under oxidative stress conditions, confirming its role in regulating ROS production. The manuscript is well written; however, some clarifications would be useful.

Manuscript Concerns:

Minor:

1. In line 17, “having both anti-inflammatory” should change to “has both anti-inflammatory”

2. In line 18, “The effects of itaconate on beta cells under oxidative stress is relatively unknown” should change to “The effects of itaconate on beta cells under oxidative stress are relatively unknown”

3. In line 20, “4-OI treatment reversed hypoxia-induced beta cell death” should change to “4-OI treatment reversed hypoxia-induced beta-cell death”.

4. In line 82, “mycoplasma contamination was regularly done by xxx kit”, please add the correct name of the kit.

5.  In lines 92,97 and 101, change the 105 per well to 105 per well.

6. Please showed all the individual dots in the bar graph.

Major:

1.  DCFH-DA is a fluorogenic dye that measures hydroxyl, peroxyl, and other reactive oxygen species (ROS) activity within the cell. The DCFDA assay protocol is based on the diffusion of DCFH-DA into the cell. It is then deacetylated by cellular esterases to a non-fluorescent compound, which is later oxidized by ROS into 2’, 7’ –dichlorofluorescein (DCF). DCF is highly fluorescent and is detected by fluorescence spectroscopy. Many groups have shown that the fluorescence should be detected in nuclear, why In Figure 2A, the fluorescence of DCFH-DA was shown in the cytoplasm?

2.  Please provide the primers and methods that how the author amplifies the LDHA construct.

3. In Figure 5D, the author pointed out metformin has an inhibitory effect on cell migration, why there are more cells in the metformin treatment group (03mM and 1mM)?

4. In Figure 4A, the western blot has shown two conditions: Hypoxia(24h) and Hypoxia(72h), but the quantification of hypoxia only shows one group?

5. In Figure 4A, the cleaved caspase-3 protein level was clearly increased in hypoxia and treatment groups compared to the normoxia group. Suggested that there is increased apoptosis in those groups (hypoxia and treatment groups). At the same time, annexin V is another apoptosis marker. Why in figure 1, there is almost no annexin V positive cells after hypoxia over 72 and 96h?

6. Please provide evidence or other references that beta cells can secret the inflammatory cytokines (IL-1b, IL-6, and TNF).

Round 2

Reviewer 1 Report

The modifications are sufficient. 

Reviewer 2 Report

The authors have answered all the questions, and the manuscript could be accepted. 

Reviewer 3 Report

Agree to accept.
